# Effectiveness of Glecaprevir/Pibrentasvir for Hepatitis C: Real-World Experience and Clinical Features of Retreatment Cases

**DOI:** 10.3390/biomedicines8040074

**Published:** 2020-04-03

**Authors:** Ayumi Sugiura, Satoru Joshita, Yuki Yamashita, Tomoo Yamazaki, Naoyuki Fujimori, Takefumi Kimura, Akihiro Matsumoto, Shuichi Wada, Hiromitsu Mori, Soichiro Shibata, Kaname Yoshizawa, Susumu Morita, Kiyoshi Furuta, Atsushi Kamijo, Akihiro Iijima, Satoko Kako, Atsushi Maruyama, Masakazu Kobayashi, Michiharu Komatsu, Makiko Matsumura, Chiharu Miyabayashi, Tetsuya Ichijo, Aki Takeuchi, Yuriko Koike, Yukio Gibo, Toshihisa Tsukadaira, Hiroyuki Inada, Yoshiyuki Nakano, Seiichi Usuda, Kendo Kiyosawa, Eiji Tanaka, Takeji Umemura

**Affiliations:** 1Department of Medicine, Division of Gastroenterology and Hepatology, Shinshu University School of Medicine, 3-1-1 Asahi, Matsumoto, Nagano 390-8621, Japan; a19860530@gmail.com (A.S.); unagichazuke.sansho.3694@gmail.com (Y.Y.); ymzktm6@shinshu-u.ac.jp (T.Y.); naoyuki.fuji@gmail.com (N.F.); kimuratakefumii@yahoo.co.jp (T.K.); amatsumo@shinshu-u.ac.jp (A.M.); tumemura@shinshu-u.ac.jp (T.U.); 2Consultation Centers for Hepatic Diseases, Shinshu University Hospital, 3-1-1 Asahi, Matsumoto, Nagano 390-8621, Japan; 3Department of Gastroenterology, Japanese Red Cross Society Nagano Hospital, 22-1 Wakasato, Nagano, Nagano 380-0928, Japan; s.wada@nagano-med.jrc.or.jp (S.W.); morihi@nagano-med.jrc.or.jp (H.M.); sshibata@shinshu-u.ac.jp (S.S.); 4Department of Gastroenterology, National Hospital Organization, Shinshu Ueda Medical Center, 27-21 Midorigaoka, Ueda, Nagano 386-8610, Japan; yoshizawa.kaname.mn@mail.hosp.go.jp (K.Y.); morita.susumu.nz@mail.hosp.go.jp (S.M.); 5Department of Gastroenterology, National Hospital Organization, Matsumoto Medical Center, 20-30 Muraimachiminami, Matsumoto, Nagano 399-8701, Japan; furuta.kiyoshi.gj@mail.hosp.go.jp (K.F.); akjojo730@ybb.ne.jp (A.K.); 6Department of Internal Medicine, Nagano Prefectural Kiso Hospital, 6613-4 Fukushima, Kiso-town, Kiso, Nagano 397-8555, Japan; iijimaakihiro_abc@yahoo.co.jp (A.I.); cha551117@yahoo.co.jp (S.K.); 7Department of Gastroenterology, Ina Central Hospital, 1313-1 Koshiroukubo, Ina, Nagano 396-8555, Japan; amaruyama@inahp.jp; 8Department of Gastroenterology, Japanese Red Cross Society Suwa Hospital, 5-11-50 Kogandori, Suwa, Nagano 392-8510, Japan; Masakazu-k@air.ocn.ne.jp (M.K.); komichi@shinshu-u.ac.jp (M.K.); 9Department of Gastroenterology, Nagano Chuo Hospital, 1570 Tsuruga-Nishitsurugamachi, Nagano, Nagano 380-0814, Japan; m-matsumura@healthcoop-nagano.or.jp; 10Department of Gastroenterology, Chikuma Central Hospital, 58 Kuiseshita, Chikuma, Nagano 387-0011, Japan; mmm@coral.ocn.ne.jp; 11Department of Gastroenterology, Japanese Red Cross Society Azumino Hospital, 5685 Toyoshina, Azumino, Nagano 399-8205, Japan; t-ichijo@azumino.jrc.or.jp; 12Aki Naika Clinic, 236-1 Nozawa, Saku, Nagano 385-0053, Japan; aki@aki-naika-clinic.jp; 13Kawanakajima Clinic, 1942-25 Kawanagajima-machi, Nagano, Nagano 381-2221, Japan; k.koike-nagano@nifty.com; 14Gibo Hepatology Clinic, 1-34-20 Muraimachiminami, Matsumoto, Nagano 399-0036, Japan; gibo@gibo-clinic.com; 15Department of Gastroenterology, Kenwakai Hospital, 1936 Kanaenakadaira, Iida, Nagano 395-8522, Japan; t-tsukadaira@kenwakai.or.jp; 16Kanebako Internal Medicine Clinic, 320-2 Kanebako, Nagano, Nagano 381-0007, Japan; curtisscw20@icloud.com; 17Nakano Gastroenterology Clinic, 4-13-5 Muraimachiminami, Matsumoto, Nagano 399-0036, Japan; qyy02452@nifty.ne.jp; 18Gastroenterology Center, Aizawa Hospital, 2-5-1 Honjo, Matsumoto, Nagano 390-0814, Japan; syoka-dr@ai-hosp.or.jp (S.U.); shinshukiyosawa@yahoo.co.jp (K.K.); 19Department of Community Medicine Promotion, Shinshu University, 3-1-1 Asahi, Matsumoto, Nagano 390-8621, Japan; etanaka@shinshu-u.ac.jp; 20Department of Life Innovation, Institute for Biomedical Sciences, Shinshu University, 3-1-1 Asahi, Matsumoto, Nagano 390-8621, Japan

**Keywords:** chronic hepatitis C, hepatitis C virus, glecaprevir, pibrentasvir, retreatment

## Abstract

Glecaprevir/pibrentasvir (G/P) are direct-acting antivirals (DAAs) that achieve a high sustained virological response (SVR) rate for hepatitis C virus (HCV) infection. We investigated G/P effectiveness for HCV patients based on real-world experience and the clinical features of retreatment cases. HCV patients (*n* = 182) were compared for clinical features and outcomes between first treatment (*n* = 159) and retreatment (*n* = 23) G/P groups. Overall, 77 patients (42.3%) were male, the median age was 68 years, and 86/66/1/4 cases had genotype 1/2/1 + 2/3, respectively. An SVR was achieved in 97.8% (178/182) of cases by intention-to-treat analysis and 99.4% (178/179) of cases by per-protocol analysis. There were no remarkable differences between the first treatment and retreatment groups for male (42.8% vs. 39.1%, *p* = 0.70), median age (68 vs. 68 years, *p* = 0.36), prior hepatocellular carcinoma (5.8% vs. 8.7%, *p* = 0.59), or the fibrosis markers AST-to-platelet ratio index (APRI) (0.5 vs. 0.5, *p* = 0.80) and fibrosis-4 (FIB-4) index (2.2 vs. 2.6, *p* = 0.59). The retreatment group had a significantly more frequent history of interferon treatment (12.3% vs. 52.2%, *p* < 0.01) and the Y93H mutation (25.0% vs. 64.7%, *p* = 0.02). The number of retreatment patients who had experienced 3, 2, and 1 DAA treatment failures was 1, 3, and 19, respectively, all of whom ultimately achieved an SVR by G/P treatment. In conclusion, G/P was effective and safe for both HCV first treatment and retreatment cases despite the retreatment group having specific resistance mutations for other prior DAAs. As G/P treatment failure has been reported for P32 deletions, clinicians should consider resistance mutations during DAA selection.

## 1. Introduction

With an estimated 130–170 million people chronically infected worldwide, including 1.5 million cases in Japan, hepatitis C virus (HCV) infection has become a global health concern, especially considering eventual disease progression to cirrhosis, hepatic failure, and hepatocellular carcinoma (HCC) [1,2,3,4]. Viral eradication is the most effective treatment to halt HCV progression. The introduction of direct-acting antiviral (DAA)-based therapies for chronic HCV infection has revolutionized HCV treatment, resulting in highly efficacious and well-tolerated therapies for nearly all patients. Recently, the effectiveness and safety of glecaprevir/pibrentasvir (G/P) was reported in a multicenter Northern Italy population, in which the treatment provided an excellent sustained virological response (SVR) rate (99.2%) with only mild adverse events [5] comparable to those in a Japanese population [6,7]. Resistance-associated substitutions (RAS) should be considered when treating patients with DAA because it was reported that 989 of 5107 patients (19.4%) had RASs in the HCV-NS5A region in a Real-World, Nationwide, Multicenter Study in Japan [8]. However, the real-world clinical impact of G/P for HCV retreatment remains unknown. We therefore examined the effectiveness and safety of G/P in a Japanese cohort with a specific focus on first treatment and retreatment cases.

## 2. Methods

### 2.1. Patients and Methods

A total of 1637 patients who had received DAA treatment and achieved 12 weeks of follow-up periods after completion of G/P treatment at Shinshu University Hospital in Matsumoto, Japan, or its affiliated hospitals, between 2015 and 2018 were initially included. Of them, 304/328 patients under daclatasvir + asunaprevir (DCV + ASV) for 24 weeks (SVR rate: 92.6%), 342/348 patients under ledipasvir/sofosbuvir (LDV/SOF) for 12 weeks (SVR rate: 98.2%), 39/44 patients under ombitasvir/paritaprevir/ritonavir (OMV/PTV/r) for 12 weeks (SVR rate: 88.6%), 80/82 patients under elbasvir + grazoprevir (EBR + GZR) for 12 weeks (SVR rate: 97.6%), and 256/264 patients under sofosbuvir + ribavirin (SOF + RBV) for 12 weeks (SVR rate: 97.0%) achieved an SVR, as shown in Figure 1. Ultimately, a total of 182 patients (median age: 68 years, 77 (42.8%) male), including 23 prior DAA failure patients, who received G/P treatment between December 2017 and November 2018 according to the Japan Society of Hepatology (JSH) clinical guidelines [9] were analyzed in this study (Table 1 and Table 2). The racial background of all patients was Japanese. The diagnosis of chronic HCV was based on JSH criteria [9] as the presence of serum HCV antibodies and detectable HCV-RNA by the real-time polymerase chain reaction (RT-PCR) at therapy initiation. Liver cirrhosis was diagnosed based on the characteristic clinical signs of advanced liver disease and/or imaging studies by attending physicians. An SVR12 was defined as undetectable HCV RNA at 12 weeks after completion of DAA therapy. Treatment failure was defined as detectable HCV RNA during treatment or within 12 weeks after completion or discontinuation of DAAs. We suspected that the reason why repeat treatment worked but the first treatment did not, was that no patient received the same DAA therapy twice. As all retreatment cases were given a different DAA (Figure 1), the initiation of differing drugs might be a point of consideration in retreatment cases. 

This study was reviewed and approved by the Institutional Review Board of Shinshu University School of Medicine (approval number: 3244) and its affiliated hospitals. Written informed consent was obtained from all participating subjects. The study was conducted according to the principals of the Declaration of Helsinki. 

### 2.2. Laboratory Testing

All laboratory data, including white blood cell count, hemoglobin, platelet count, albumin, aspartate aminotransferase (AST), alanine aminotransferase (ALT), and alpha fetoprotein, were determined using standard methods at respective institutions. The fibrosis-4 (FIB-4) index and AST-to-platelet ratio index (APRI) were respectively calculated as: age (years) × AST (U/L)/platelet count (x109/L) × ALT (U/L)1/2 [10] and (AST/upper limit of normal; 40 (U/L)) × (100/platelet count (109/L)) [11].

### 2.3. Testing for RAS of the HCV-RNA Genome

Among 159 patients who were treated with G/P as the first DAA treatment, 19 were analyzed for the NS5A-Y93H RAS mutation by RT-PCR prior to G/P treatment as described previously [12], with a value of 20% or more defined as NS5A-Y93H positivity [13] in a commercially based laboratory [14]. All retreatment cases with HCV genotype 1, except for 3 with a failed combination of interferon (IFN) and DAA treatment, were analyzed for RAS in the NS5A and NS3 regions (Table 3). All cases with HCV genotype 2 were analyzed for RAS in NS5B (Table 4) by a direct sequencing method.

### 2.4. Statistical Analysis

Statistical analysis and data visualization were carried out using StatFlex version 7.0.7 (Artech Co., Ltd., Osaka, Japan). Continuous baseline data are expressed as the median ± interquartile range and statistically evaluated by means of the Mann–Whitney U test. Categorical variables are presented as the frequency (percentage) and analyzed using the chi-square test. All statistical tests were two-sided and evaluated at the 0.05 level of significance.

## 3. Results

### 3.1. Baseline Clinical Characteristics

The baseline clinical characteristics in this study are summarized in Table 1. Of the 182 enrolled patients, 77 (42.3%) were male, 105 (57.7%) were female, and the median age was 68 years. The number of patients at the chronic hepatitis stage or liver cirrhosis stage was 134 and 25 cases, respectively. Patients infected with HCV genotype 1, 2, 1 + 2, or 3 numbered 86, 66, 1, and 4, respectively, as shown in Table 1. Roughly half of the patients had the Y93 mutation in the NS5A region. Hypertension was the most frequent complication (51 patients; 37%), followed next by diabetes mellitus in 20 patients (16%), dyslipidemia in 12 patients (8.7%), and renal failure under hemodialysis in eight patients (5.0%). Twenty-nine patients (18%) had interferon (IFN) treatment experience, and 10 patients (6.2%) had a history of HCC. Overall, 23 patients (13.6%) were prior DAA treatment failure cases (retreatment group), which included three cases of triple therapy (simeprevir or vaniprevir/pegylated IFN/ribavirin), seven cases of DCV/ASV, three cases of LDV/SOF, one case of EBR + GZR, one case of OMV/PTV/r, four cases of SOF + RBV, two cases of DCV/ASV followed by LDV/SOF, one case of DCV/ASV followed by EBR + GRZ, and one case of SMV followed by DCV/ASV followed by LDV/SOF.

### 3.2. G/P Outcomes

In total, an SVR was achieved in 97.8% (178/182) of cases by intention-to-treat analysis and 99.4% (178/179) of cases by per-protocol analysis. One (0.5%) treatment-naïve, non-cirrhotic stage, genotype 2a-infected male patient experienced a virological relapse at eight weeks of follow-up. In the retreatment group, the number of patients who had experienced 3, 2, and 1 DAA treatment failures was 1, 3, and 19, respectively, all of whom ultimately achieved an SVR by G/P treatment. 

Of 182 patients treated with G/P, only three discontinued treatment, which supported previous studies on the safety of G/P therapy [5,6]. Specifically, one female patient (0.5%) discontinued treatment due to the adverse event of diarrhea 2 weeks after commencing G/P. Her HCV-RNA was negative 4 weeks after starting treatment but had become positive at 12 weeks. Another female patient (0.5%) elected to stop therapy for personal reasons despite negativity for HCV-RNA at 4 weeks. A male patient (0.5%) died of cerebral infarction unrelated to G/P treatment. His HCV-RNA had been negative after 4 weeks of drug administration. Although patients with HCV genotype 3 have been reported to achieve comparatively lower SVR rates [15], all such patients (*n* = 4) in this study attained an SVR by G/P, thus confirming a previous study demonstrating high SVR rates regardless of genotype [16]. 

### 3.3. Comparisons between First Treatment and Retreatment Groups

Comparisons of the clinical features and outcomes between the first treatment (*n* = 159) and retreatment (*n* = 23) G/P therapy groups are presented in Table 2. There were no remarkable differences between the groups for male gender frequency (43.3% vs. 39.1%, *p* = 0.705), median age (68 vs. 68 years, *p* = 0.362), frequency of cirrhosis stage (17.4% vs. 9.1%, *p* = 0.357), frequency of prior HCC history (5.8% vs. 8.7%, *p* = 0.593), or the liver fibrosis markers of APRI (0.5 vs. 0.5, *p* = 0.805) or FIB-4 index (2.2 vs. 2.6, *p* = 0.592). However, the retreatment group had a significantly higher frequency of IFN treatment history (12.3% vs. 52.2%, *p* < 0.001) and the Y93H mutation (25.0% vs. 75.0%, *p* < 0.001). Y93H was the most frequent mutation in the retreatment group (Table 3). A variety of RASs were detected in the NS5A, NS3, and NS5B regions, but G/P could successfully achieve an SVR in all cases.

### 3.4. RASs in HCV-RNA Genome in Retreatment Cases

As shown in Table 3, retreatment cases in genotype 1 had a variety of types of RASs in NS5A while those in genotype 2 had no RASs in NS5B. 

## 4. Discussion

This study presents important evidence on G/P treatment efficacy and safety for HCV eradication in first and retreatment cases based on our real-world experience.

In the cohort, one case with HCV genotype 2a had unsuccessful G/P treatment as the first DAA therapy. RAS had not been analyzed before G/P, but later testing revealed none in the NS5B region. Another case was previously reported as a failed 8-week course of G/P therapy [6], which suggested that 8 weeks of treatment was insufficient to achieve an SVR for genotype 2a HCV, as evidenced in the present series. 

It was noteworthy that eight patients receiving hemodialysis achieved an SVR under G/P, which supported previous reports that even individuals with accompanying conditions could achieve an SVR safely and effectively by a G/P regimen [17]. We very recently described that a history of HCC was an independent risk factor of DAA treatment failure [18]. However, all 10 patients (6.2%) with prior HCC in our cohort achieved an SVR to further highlight the broad utility of G/P. Unfortunately, six of 49 DAA treatment failure patients were unable to receive further treatment, including G/P, due to complicating HCC after DAA failure, suggesting the need for earlier intervention. 

G/P is the first pan-genotype DAA treatment for HCV in the world. Although genotype 3 patients are a minority in Japan, genotype 3 HCV has been resistant to SOF + RBV regimens, with an SVR rate of approximately 35% [19]. In Japan, a DAA-based regimen of SOF + RBV for 24 weeks was approved for genotype 3 in 2017, which could achieve an SVR rate of 85% [20]. Our study included four patients with this genotype, all of whom achieved an SVR. Genotype 3 HCV was reported to show relatively rapid liver fibrosis progression [21]. Therefore, G/P may be a good option for treatment before advancement to liver fibrosis. 

The retreatment group had a significantly higher frequency of IFN treatment history and the Y93H mutation, suggesting a resistance to prior IFN therapy and the acquisition of a resistance mutation following prior DAA treatment. Although the Y93H mutation on NS5A has been associated with DAA treatment failure [22], all 12 patients with this mutation achieved an SVR, indicating that it was unrelated to G/P outcome. The P32 deletion has also been strongly linked to G/P therapy ineffectiveness by previous clinical reports [23,24], but we were unable to conclude a relationship with G/P failure as there was only one P32 deletion patient in our cohort who had acquired the mutation after DCV/ASV failure and did not receive G/P due to prior identification of the deletion. Regarding the administration of DCV + ASV, if patients harbored the NS5A-Y93H RAS and could no longer postpone treatment due to age, disease progression, or other clinical reasons, they commenced the regimen instead of waiting for next-generation DAAs, which could achieve a high SVR rate of 91.7% as reported previously [18]. The relatively low frequency of the P32 deletion in our cohort was presumably due to the above strategy. It has also been confirmed by basic research evidence that a P32 deletion conferred severe drug resistance even to second-generation NS5A inhibitors [25], thus supporting a strategy of resistance mutation testing before treatment to halt the creation of new unidentified mutations in the HCV genome. Taken together, our data confirmed that G/P was effective not only for HCV first treatment, but also for retreatment, despite the retreatment group having specific resistance mutations to other DAAs. 

There are several limitations to this study. It was retrospectively conducted, and the assessed data on safety and tolerability were based on medical records by attending physicians. Additionally, RASs measurement was limited in some cases because it is not covered by the health insurance system in our country. G/P efficacy and safety should also be analyzed in other ethnicities to confirm the results of the present study. 

## 5. Conclusions

G/P treatment appears effective and safe for patients with HCV, regardless of DAA treatment history, which highlights the value of a micro-elimination approach to global HCV eradication. As G/P treatment failure has been reported for P32 deletions, clinicians should consider resistance mutations during DAA selection.

## Figures and Tables

**Figure 1 biomedicines-08-00074-f001:**
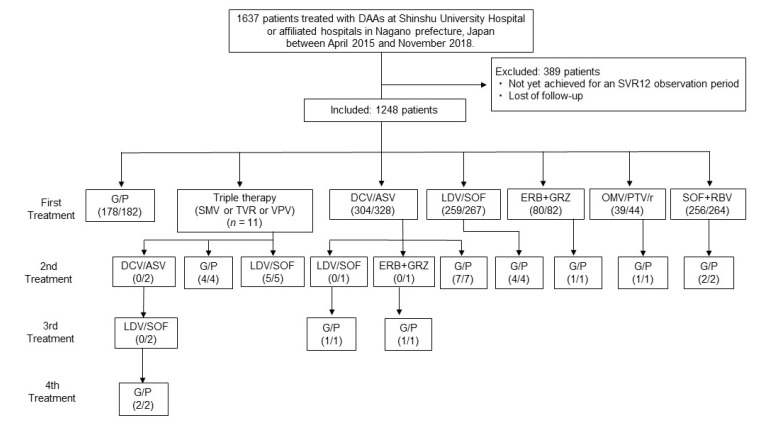
Study flowchart showing the efficacy of direct-acting antiviral (DAA) treatment at SVR12. The respective number of patients who achieved a sustained virological response (SVR) and received DAA treatment is shown. Abbreviations: DAA, direct-acting antiviral; SVR, sustained virological response; G/P, glecaprevir/pibrentasvir; SMV, simeprevir; TVR, telaprevir; VPV, vaniprevir; DCV + ASV, daclatasvir + asunaprevir; LDV/SOF, ledipasvir/sofosbuvir; OMV/PTV/r, ombitasvir/paritaprevir/ritonavir; EBR + GRZ, elbasvir + grazoprevir; SOF + RBV, sofosbuvir + ribavirin.

**Table 1 biomedicines-08-00074-t001:** Baseline patient characteristics.

	All Patients (*n* = 182)	
	Median	IQR
Age at entry (years)	68	(18–93)
Gender (male/female)	77/105	
HCV genotype		
1/2/3/1 + 2/ND	86/66/4/1/25	
Chronic hepatitis/liver cirrhosis	155/27	
Laboratory data		
White blood cells (μ/L)	4910	(1360–9010)
Hemoglobin (g/dL)	13.7	(8.7–18.5)
Platelet count (×10^9^/L)	176	(12–369)
Albumin (g/dL)	4.1	(2.7–4.8)
AST (U/L)	33	(11–259)
ALT (U/L)	29	(7–281)
Total bilirubin (mg/dL)	0.7	(0.2–2.0)
AFP (ng/mL)	4.0	(1.2–193.0)
Cre (mg/dL)	0.7	(0.4–11.4)
eGFR (mL/min/1.73 m^2^)	68.1	(3.0–118.1)
Fibrosis markers		
FIB-4 index	2.3	(0.3–27.2)
APRI	0.5	(0.1–17.5)
Y93H mutation *	50.0%	
Complications		
Hypertension	37.0%	
Diabetes	15.6%	
Dyslipidemia	8.7%	
Hemodialysis	5.0%	
Experienced		
Prior IFN	18.0%	
Prior DAAs	13.6%	
Prior HCC	6.2%	
Outcome		
SVR (ITT)	178/182 (97.8%)	
SVR (PP)	178/179 (99.4%)	
Discontinuation	3/182 (0.16%)	

*: Of 159 G/P first treatment cases, 20 were analyzed for a NS5A-Y93H resistance-associated substitution (RAS) mutation by RT-PCR. Of 23 retreatment cases, 16 genotype 1 cases were analyzed for a RAS mutation by a direct sequencing method. Abbreviations: G/P, glecaprevir/pibrentasvir; IQR, interquartile range; HCV, hepatitis C virus; ND, not determined; AST, aspartate aminotransferase; ALT, alanine aminotransferase; AFP, alpha fetoprotein; NA, not applicable; Cre, creatinine; eGFR, estimated glomerular filtration rate; APRI, aspirate aminotransferase-to-platelet ratio index; IFN, interferon; DAAs, direct acting antivirals; HCC, hepatocellular carcinoma; ITT, intention-to-treat analysis; PP, per-protocol analysis.

**Table 2 biomedicines-08-00074-t002:** Comparison of clinical characteristics between first treatment and retreatment cases.

	First Treatment (*n* = 159)	Retreatment (*n* = 23)	*p*-Value
	Median (IQR)	Median (IQR)	
Age at entry (years)	68	68	0.362
Gender (male/female)	68/91	9/14	0.705
HCV genotype			
1/2/3/1 + 2/ND	66/63/4/1/25	19/4/0/0/0	
Chronic hepatitis/liver cirrhosis	135/24	20/3	0.357

Laboratory data			
White blood cells (μ/L)	4900 (1360–9010)	4920 (2310–8670)	0.937
Hemoglobin (g/dL)	13.5 (8.7–17.6)	14.3 (8.7–18.5)	0.111
Platelet count (×10^9^/L)	176 (12–369)	174 (34–343)	0.249
Albumin (g/dL)	4.1 (2.7–4.8)	4.1 (2.8–4.7)	0.653
AST (U/L)	33 (11–259)	33 (19–90)	0.580
ALT (U/L)	30 (7–281)	27 (15–141)	0.458
Total bilirubin (mg/dL)	0.6 (0.2–2.0)	0.7 (0.4–1.5)	0.399
AFP (ng/mL)	4.0 (1.2–193.0)	5.0 (1.7–24.6)	0.455
Cre (mg/dL)	0.8 (0.5–11.1)	0.7 (0.5–3.0)	0.180
eGFR (mL/min/1.73 m^2^)	66.9 (3.0–115.7)	78.3 (17.8–118.1)	0.061
Fibrosis markers			
FIB-4 index	2.2 (0.3–27.3)	2.6 (0.8–9.7)	0.592
APRI	0.5 (0.1–17.5)	0.5 (0.1–3.8)	0.805
Y93H mutation *	25.0%	75.0%	<0.001
Complications			
Hypertension	37.0%	36.8%	0.991
Diabetes	17.4%	5.0%	0.158
Dyslipidemia	9.2%	5.3%	0.567
Hemodialysis	5.8%	0%	0.236
Experienced			
Prior IFN	12.3%	52.2%	<0.001
Prior HCC	5.8%	8.7%	0.593

*: Of 159 G/P first treatment cases, 20 were analyzed for a NS5A-Y93H RAS mutation by RT-PCR. Of 23 retreatment cases, 16 genotype 1 cases were analyzed for a NS5A-Y93H RAS mutation by a direct sequencing method. Abbreviations: G/P, glecaprevir/pibrentasvir; IQR, interquartile range; HCV, hepatitis C virus; ND, not determined; AST, aspartate aminotransferase; ALT, alanine aminotransferase; AFP, alpha fetoprotein; NA, not applicable; Cre, creatinine; eGFR, estimated glomerular filtration rate; APRI, aspirate aminotransferase-to-platelet ratio index; IFN, interferon; DAAs, direct acting antivirals; HCC, hepatocellular carcinoma.

**Table 3 biomedicines-08-00074-t003:** Resistance-associated substitutions of the HCV-RNA genome in genotype 1 retreatment cases.

Case	Age	Gender	Prior DAA Treatment	NS5A												NS3										
Genotype 1				L23	Q24	L28	R30	L31	P32	F37	Q54	P58	Q62	A92	Y93	V36	F43	T54	V55	N77	Q80	S122	R155	A156	D168	V170
1	55	F	DCV/ASV	-	-	-	-	M	-	-	-	S	-	-	H	-	-	-	-	-	-	-	-	-	-	-
2	69	M	DCV/ASV	-	K	M	R/A/G/T	-	-	-	-	-	-	-	-	-	-	-	-	-	L	G	-	-	-	-
3	74	F	DCV/ASV	-	-	-	-	F	-	-	H	-	-	V	H	-	-	-	-	-	-	G	-	-	-	-
4	75	F	DCV/ASV	-	K	T	H	-	-	-	-	-	-	-	-	-	-	-	-	-	-	G	-	-	-	I
5	34	F	DCV/ASV	-	-	-	-	V	-	L	G	-	-	-	H	-	-	-	-	-	L	-	-	-	E	I
6	68	F	DCV/ASV	-	K	M/V	H/Q	-	-	-	-	-	-	-	Y/H	-	-	-	-	-	Q/R	-	-	-	E	I
7	70	F	DCV/ASV	-	-	-	-	-	-	-	Y	-	-	-	H	-	-	-	-	-	-	G	-	-	-	-
8	76	F	LDV/SOF	-	-	-	-	L/M	-	-	-	-	-	-	-	-	-	-	-	-	-	-	-	-	-	I
9	37	M	LDV/SOF	-	K	M	Q	-	-	L	N	-	-	-	H/N	-	-	-	-	-	-	S/N	-	-	-	I
10	70	F	LDV/SOF	-	-	-	-	-	-	F/L	-	-	-	-	H	-	-	-	-	-	-	N	-	-	-	-
11	68	F	EBR + GRZ	-	-	-	-	V	-	L	H	-	-	-	H	-	-	-	-	-	-	-	-	-	-	-
12	65	F	OBV/PTV/r	-	-	-	-	-	-	-	H	-	D	-	Y/H	-	-	-	-	-	-	T	-	-	-	I
13	76	M	DCV/ASV followed by LDV/SOF	-	K	M	Q	-	-	L	R	-	E	-	H	-	-	-	-	-	-	-	-	-	-	I
14	58	M	DCV/ASV followed by LDV/SOF	-	-	-	-	V	-	L	-	-	-	-	H	-	-	-	-	-	L	C	-	-	E	I
15	82	M	DCV/ASV followed by EBR + GRZ	-	K	M	Q	V	-	-	-	-	-	-	F	-	-	-	-	-	-	-	-	-	-	-
16	63	M	SMV followed by DCV/ASV followed by LDV/SOF	-	-	-	-	I/M	-	L	H	-	-	-	Y/H	-	-	-	-	-	L	-	-	-	D/E	I
17 *	28	M	VPV	NT	NT	NT	NT	NT	NT	NT	NT	NT	NT	NT	NT	NT	NT	NT	NT	NT	NT	NT	NT	NT	NT	NT
18 *	70	F	SMV	NT	NT	NT	NT	NT	NT	NT	NT	NT	NT	NT	NT	NT	NT	NT	NT	NT	NT	NT	NT	NT	NT	NT
19 *	57	F	SMV	NT	NT	NT	NT	NT	NT	NT	NT	NT	NT	NT	NT	NT	NT	NT	NT	NT	NT	NT	NT	NT	NT	NT
Major RAS				L23	Q24K	L28M	R30Q	L31M	P32	F37L	Q54H	P58S	Q62	A92V	Y93H	V36	F43	T54	V55	N77	Q80L	S122G	R155	A156	D168E	V170I
Frequency				0	37.5%	31.2%	25.0%	18.7%	0	43.7%	25.0%	6.2%	0	6.2%	75.0%	0	0	0	0	0	25.0%	25.0%	0	0	25.0%	56.2%

*: Cases #17–19 who had failed in VPV and SMV regimens did not tested because they did not need for RAS measurement for health insurance system inclusion at that time. Abbreviations: RAS, resistance-associated substitution; -, no RAS; NT, not tested; DCV/ASV, daclatasvir + asunaprevir; LDV/SOF, ledipasvir/sofosbuvir; EBR + GZR, elbasvir + grazoprevir; OMV/PTV/r, ombitasvir/paritaprevir/ritonavir; SOF + RBV, sofosbuvir + ribavirin; SMV, simeprevir; VPV, vaniprevir.

**Table 4 biomedicines-08-00074-t004:** Resistance-associated substitutions of the HCV-RNA genome in genotype 2 retreatment cases and a G/P failure case.

Case	Age	Gender	Prior DAA Treatment	NS5B				
Genotype 2				L159	S282	C316	L320	V321
20	59	M	SOF + RBV	-	-	-	-	-
21	50	M	SOF + RBV	-	-	-	-	-
22	77	F	SOF + RBV	-	-	-	-	-
23	53	M	SOF + RBV	-	-	-	-	-
Major RAS				L159	S282	C316N	L320	V321
Frequency				0	0	0	0	0
G/P failure case	75	M	Before G/P	NT	NT	NT	NT	NT
			After G/P	-	-	-	-	-

Abbreviations: RAS, resistance-associated substitution; NT, not tested; DCV/ASV, daclatasvir + asunaprevir; LDV/SOF, ledipasvir/sofosbuvir; EBR + GZR, elbasvir + grazoprevir; OMV/PTV/r, ombitasvir/paritaprevir/ritonavir; SOF + RBV, sofosbuvir + ribavirin; SMV, simeprevir; VPV, vaniprevir.

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
