# Peer review of "Effectiveness of Glecaprevir/Pibrentasvir for Hepatitis C: Real-World Experience and Clinical Features of Retreatment Cases"

_biomedicines, 2020, doi:10.3390/biomedicines8040074_

Round 1
Reviewer 1 Report
Manuscript “Effectiveness of glecaprevir/pibrentasvir for hepatitis C: Real-world experience and clinical features of retreatment cases” by Sugiura et al., provides important information for clinical application of glecaprevir/pibrentasvir combination and highlight the importance of such regiments for retreatment of patients with existing resistant mutations. The manuscript is well written, easy to read and will be useful for the world-wide treatment of HCV.
Minor comment: Reference #7 needs to be completed.
Author Response
Thank you for giving us the opportunity to resubmit our manuscript, entitled “Effectiveness of glecaprevir/pibrentasvir for hepatitis C: Real-world experience and clinical features of retreatment cases”, for publication in Biomedicines. Your comments and those of the Reviewers have been helpful in allowing us to revise our report. All additions and changes are highlighted in the resubmitted version. We have attempted to address the questions raised by the Reviewers as follows:
Response to Reviewer 1:
- Minor comment: Reference #7 needs to be completed.
Thank you for pointing this out. The reference has been completed. (page 11, line 307)

Reviewer 2 Report
This manuscript provides an analysis of hepatitis C patients who underwent Glecaprevir/pibrentasvir (G/P) treatment. It showed that the G/P treatment achieved more than 97% success rate (SVR). Most importantly, all of those failure cases succeeded to achieve SVR after the re-treatment. These results confirmed the previous reports in the literature (ref. 5-7 in the original manuscript). Thus, the G/P treatment should be the treatment of choice in these situations even when the patients have other mutations not related to the G/P sequence. Such information is useful but not original. As such, the manuscript can be made more succinct by not showing the details of the patients.
Specific suggestion
- Since the results showed that the initial SVR patients and non-SVR have similar responses in the first and second treatments, there is no need to show the demographical and clinical data of these patients (Table 1 and 2).
- Please comment on why the repeat treatment worked but the first treatment did not.Is it just the length or dose of the treatment?
Author Response
Thank you for giving us the opportunity to resubmit our manuscript, entitled “Effectiveness of glecaprevir/pibrentasvir for hepatitis C: Real-world experience and clinical features of retreatment cases”, for publication in Biomedicines. Your comments and those of the Reviewers have been helpful in allowing us to revise our report. All additions and changes are highlighted in the resubmitted version. We have attempted to address the questions raised by the Reviewers as follows:
Responses to Reviewer 2:
- Since the results showed that the initial SVR patients and non-SVR have similar responses in the first and second treatments, there is no need to show the demographical and clinical data of these patients (Table 1 and 2).
We appreciate this comment. As the reviewer points out, there are no significant differences between the first treatment and retreatment therapy between the groups except for the frequency of IFN treatment history and the Y93H mutation in Table 2. From these findings, we would like to establish two main messages, as follows: 1) G/P was effective not only for HCV first treatment, but also for retreatment, and 2) a strategy of resistance mutation testing before treatment should be considered to halt the creation of new unidentified mutations in the HCV genome. Therefore, we would like to retain Tables 1 and 2 in the report. We have added the above points to the discussion section. (page 10, line 252)
- Please comment on why the repeat treatment worked but the first treatment did not. Is it just the length or dose of the treatment?
Thank you for this question. We suspected that the reason why repeat treatment worked but the first treatment did not was that no patient received the same DAA therapy twice. As all retreatment cases were given a different DAA (Figure 1), the initiation of differing drugs might be a point of consideration in retreatment cases. (page 3, line 113)

Reviewer 3 Report
Herewith please find the comments for the manuscript titled "Effectiveness of glecaprevir/pibrentasvir for hepatitis C: Real-world experience and clinical features of retreatment cases"
- The introduction section is too short and not specific. The authors should explain their country background related to HCV RAS frequency and should address important issues related to it.
- In the methods section RAS testing was not clear. The authors though shown reference for this methods should atleast explain some information on RAS testing.
- Among 159 patients who were treated with G/P as the first DAA treatment why authors tested only 20 patients? On what basis these samples were selected?
- In line number 142-143 stated 20 were analyzed for the NS5A-Y93H RAS mutation by RT-PCR but in table only showed for 19 patients?
- Are these sequences from these study submitted to genbank? If so include the genbank number.
- Why patients 17 to 19 not tested for RAS? It was not addressed in the text.
- The results section should be improved drastically. There was no information about RAS in this results section.
- The conclusion is too short and not specific.
- Overall, the authors should improve message on all sections.
Author Response
Thank you for giving us the opportunity to resubmit our manuscript, entitled “Effectiveness of glecaprevir/pibrentasvir for hepatitis C: Real-world experience and clinical features of retreatment cases”, for publication in Biomedicines. Your comments and those of the Reviewers have been helpful in allowing us to revise our report. All additions and changes are highlighted in the resubmitted version. We have attempted to address the questions raised by the Reviewers as follows:
Responses to Reviewer 3:
- The introduction section is too short and not specific. The authors should explain their country background related to HCV RAS frequency and should address important issues related to it.
Thank you for this suggestion. It was recently reported that 989 of 5,107 cases had a RAS in the HCV-NS5A region (Toyoda H, et al. Open Forum Infect Dis 2019, 6, (5), ofz185.). We have cited this reference and addressed this point in the introduction section.
- In the methods section RAS testing was not clear. The authors though shown reference for this method should at least explain some information on RAS testing.
We appreciate this suggestion. RAS was tested at a commercially based laboratory. We have added this detail in the methods section and cited an appropriate reference (Miura M, et al. Hepatol. Res. 2014, 44, (14), E360-7).
- Among 159 patients who were treated with G/P as the first DAA treatment why authors tested only 20 patients? On what basis these samples were selected?
Thank you for this question. RAS measurement is not covered by the national health insurance system of our country, and the guidelines from the Japan Society of Hepatology do not recommend RAS testing for all cases because G/P therapy has no inferiority if patients have RASs apart from a P32 deletion. Therefore, few cases were tested for a RAS in this study. We have addressed this point as a study limitation.
- In line number 142-143 stated 20 were analyzed for the NS5A-Y93H RAS mutation by RT-PCR but in table only showed for 19 patients?
Thank you for pointing this out. We have corrected the oversight. (page 5, line 48)
- Are these sequences from this study submitted to genbank? If so include the genbank number.
We have not submitted the data to genbank.
- Why patients 17 to 19 not tested for RAS? It was not addressed in the text.
Thank you for raising this point. All cases that receive DAA retreatment must undergo RAS testing for coverage by the national health insurance system of our country, except for those with failure of first generation DAAs, such as VPV or SMV combination with IFN. Therefore, patients 17-19 did not receive RAS evaluation prior to starting DAA retreatment. We have explained this in the Table 3 legend. (page 7, line 157)
- The results section should be improved drastically. There was no information about RAS in this results section.
Thank you for this comment. We have added more information on RAS findings in the results section. (page 9, line 214)
- The conclusion is too short and not specific.
Thank you for pointing this out. We have reworded the conclusion. (page 10, line 263)

Round 2
Reviewer 2 Report
The revision is satisfactory.
A typo: line 157, "did not tested" should be "were not tested"